# Radiomics Analysis of [^18^F]FDG PET/CT Thyroid Incidentalomas: How Can It Improve Patients’ Clinical Management? A Systematic Review from the Literature

**DOI:** 10.3390/diagnostics12020471

**Published:** 2022-02-12

**Authors:** Mirela Gherghe, Alexandra Maria Lazar, Mario-Demian Mutuleanu, Adina Elena Stanciu, Sorina Martin

**Affiliations:** 1Nuclear Medicine Department, University of Medicine and Pharmacy Carol Davila Bucharest, 050474 Bucharest, Romania; mirela.gherghe@umfcd.ro (M.G.); mario-demian.mutuleanu@drd.umfcd.ro (M.-D.M.); 2Nuclear Medicine Department, Institute of Oncology “Profesor Doctor Alexandru Trestioreanu”, 022328 Bucharest, Romania; 3Carcinogenesis and Molecular Biology Department, Institute of Oncology “Profesor Doctor Alexandru Trestioreanu”, 022328 Bucharest, Romania; adinaelenastanciu@yahoo.com; 4Endocrinology Department, Elias Emergency University Clinic Hospital, 011461 Bucharest, Romania; sorina.martin@gmail.com; 5Endocrinology Department, University of Medicine and Pharmacy Carol Davila Bucharest, 050474 Bucharest, Romania

**Keywords:** thyroid incidentaloma, radiomics, [^18^F]FDG PET/CT, volumetric parameters

## Abstract

Background: We performed a systematic review of the literature to provide an overview of the application of PET-based radiomics of [^18^F]FDG-avid thyroid incidentalomas and to discuss the additional value of PET volumetric parameters and radiomic features over clinical data. Methods: The most relevant databases were explored using an algorithm constructed based on a combination of terms related to our subject and English-language articles published until October 2021 were considered. Among the 247 identified articles, 19 studies were screened for eligibility and 11 met the criteria, with 4 studies including radiomics analyses. Results: We summarized the available literature based on a study of PET volumetric parameters and radiomics. Ten articles provided accurate details about volumetric parameters and their prospective value in tumour assessment. We included the data provided by these articles in a sub-analysis, but could not obtain statistically relevant results. Four publications analyzed the diagnostic potential of radiomics and the possibility of creating precise predictive models, their corresponding quality score being assessed. Conclusions: The use of PET volumetric parameters and radiomics analysis in patients with [^18^F]FDG-avid thyroid incidentalomas outlines a great prospect in diagnosis and stratification of patients with malignant nodules and may represent a way of limiting the need for unnecessary invasive procedures; however, further studies need to be performed for a standardization of the method.

## 1. Introduction

Thyroid incidentaloma (TI) is defined as a thyroid gland lesion incidentally and newly discovered during non-thyroid related imagistic examinations, such as ultrasound, conventional radiology, and nuclear medicine investigations, performed for unrelated and non-thyroid purposes [1,2]. Incidentally discovered [^18^F]-Fluorodeoxyglucose (FDG)-avid thyroid lesions have been reported in about 1.2–4.3% [2,3,4,5,6,7]; between 14–63.3% have been confirmed as malignant [2,3,4,5,6,7,8,9], with some authors stating that the malignancy rate could reach up to 80% [7]. However, the great statistical heterogeneity between studies must be taken into consideration [2,9,10]. One of the most extensive reviews on this matter, Bertagna et al., states that the average of malignant lesions found in patients with thyroid incidentalomas is formed by approximatively one-third of the patients who underwent further investigations for their incidental findings [2]. Given the limited understanding of the tumour’s natural history, a need for differentiation between malignant and benign lesions has been expressed, in order to avoid more invasive procedures, such as fine-needle aspiration (FNA) or surgical biopsy.

Over the past few years, there has been a growing interest in creating a non-invasive predicting model for discriminating malignant from benign nodules based on their [^18^F]FDG uptake on PET/CT studies. Radiomics is a rapidly developing field of research that has, as a common objective in oncology, the characterization of the tumour phenotype, which can be achieved by quantitative measurements of a high number of features, designed to capture specific characteristics of an image [11]. Radiomics implies the extraction of features to characterize a volume of interest (VOI) in the images. These features can be classified into different categories: (a) histogram-based features, (b) texture-based features, (c) edges features, and (d) shape features [12]. Accurate PET-derived criteria could help stratify the risk of malignancy in [^18^F]FDG-avid TIs, reducing the need for further evaluation and other associated risks, costs, and patient discomfort [13]. Studies have shown that combining PET-derived conventional parameters, such as standardized uptake value (SUV), metabolic tumour volume (MV), and total lesion glycolysis (TLG), with radiomics features may improve diagnostic accuracy and reduce the need for invasive diagnostic procedures [14]. Furthermore, texture analysis and machine learning approaches applied to medical imaging may be of value in the future for identifying and characterizing thyroid nodules, and easily stratifying the ones with aggressive behaviour [14,15,16].

Our report provides a qualitative review of the existing literature, describing the applied radiomics methods in patients with incidental thyroid uptake on [^18^F]FDG-PET/CT.

## 2. Materials and Methods

### 2.1. Literature Search

A comprehensive search algorithm in the PubMed/MEDLINE databases was constructed based on the combination of the following terms: “thyroid incidentaloma”, “[^18^F]FDG PET/CT”, “volumetric parameters”, “radiomics”, “thyroid texture analysis”. No beginning date was applied and the search extended up to October 2021. To expand our research, we manually evaluated the references from the retrieved articles to forge for supplementary useful studies. We followed the Preferred Reporting Items for Systematic Reviews and Meta-Analysis (PRISMA) guidelines to select the relevant studies [17] (Figure 1). All studies or subsets in studies investigating the role of either PET/CT volumetric parameters and/or radiomics in patients with [^18^F]FDG-avid thyroid incidentaloma on PET/CT were considered eligible for inclusion. Two authors performed an initial screening of identified titles and abstracts.

Our review is aimed at describing the utility of a correlation between PET/CT volumetric parameters and radiomics features, with the objective of creating an algorithm that could be introduced in routine medical practice and may have an impact on patient management in the near future; thus, we focused our research on studies performed in clinical settings.

The exclusion criteria were: (a) articles not within the field of interest; (b) articles written in other languages than English; (c) case reports or small case series; (d) in vitro or animal studies; (e) reviews and meta-analysis articles, letters, comments, or conference proceedings.

Among the 247 articles identified after the first search, 19 studies were assessed for eligibility, and 11 studies were selected and used for this qualitative synthesis. The studies were grouped into two different sets, based on the types of PET/CT parameters analysed by each research group. Data extracted from each publication included authors, research design, study reasons, imaging techniques, number of patients studied, number of patients with incidental thyroid findings, SUV_max_ values, volumetric parameters values, radiomics analysis technique, and radiomics results. 

In the following review, several specific PET parameters and textural features will be named. For the mathematical definition and further description of each of these terms, we refer readers to the original publications, with the caveat that different names might be applied to the same element, due to the lack of a standardization of terms caused by the novelty of this branch of medical imaging.

### 2.2. Data Synthesis

Data synthesis was performed using IBM SPSS Statistics Version 26 (IBM, SPSS, Inc., Chicago, IL, USA, 2019) and Microsoft Excel 2021. The total number of patients with malignant thyroid incidentalomas was obtained by adding the patients from all the publications. A sub-analysis was performed in the ten studies that reported precise data of their results, and thereby allowed calculations of the overall mean SUV_max_, MTV and TLG for the malignant lesions. Means and standard deviation (SD) were calculated using the Quantile Estimation method developed by McGrath et al. [18] (https://smcgrath.shinyapps.io/estmeansd/, last accessed on 12 January 2022), when the author provided the median value and minimum and maximum values. One sample T-Test was used to evaluate the statistical significance of the values obtained for mean SUV_max_, MTV, and TLG; a *p* < 0.05 was considered as valid. To assess the overall quality of the considered radiomics features, the radiomics quality score RQS metric described by Lambin and colleagues [19] was adopted and a score was calculated for each paper that included radiomic analysis.

## 3. Results

### 3.1. Diagnostic Value of PET/CT Volumetric Parameters for Characterization of FDG-Avid Thyroid Incidentaloma

Table 1 summarizes the main results of the studies focused on research on PET/CT volumetric parameters in the assessment of [^18^F]FDG-avid thyroid incidentaloma. The majority of papers have aimed to establish the diagnostic value of the most widely used PET/CT parameters in differentiating between benign and malignant thyroid incidental findings.

Most past studies usually focused on the discriminating power of SUV values in distinguishing between malignant and benign incidental thyroid nodules. However, their results were inconclusive and the need for further metabolic parameters as diagnostic tools arose.

The first study to analyse the usage of other PET/CT parameters as diagnostic tools was developed by Kim B.H. et al. [20] from 2010 to 2012, specifically researching the diagnostic value of volume-based parameters and SUV_max_ using ^18^F-FDG PET/CT for differentiation between malignant and benign thyroid incidentaloma in 249 patients with 262 nodules, of whom 177 underwent fine-needle aspiration biopsy (FNAB). The study had a prevalence of malignancy of 20.9% and showed that SUV_max_ and MTV at a cut-off value of 4 for SUV_max_ (MTV4) measured by ^18^F-FDG PET/CT were able to predict malignancy and could improve diagnostic accuracy (area under the curve, AUC = 0.655, respectively AUC = 0.650).

The same research group also investigated the predictive value of MTV and TLG in detecting lateral lymph node metastases (LNM) in patients with ^18^F-FDG thyroid incidentaloma [21]. Out of the 358 patients with FDG-avid thyroid nodules, 235 were subjected to FNAB, and 51 underwent total thyroidectomy with lymph node dissection and represented the sample for this study. The results showed that the SUV_max_, MTV and TLG measured by ^18^F-FDG PET/CT were significantly associated with lateral LNM in patients with incidentally detected malignant nodules, with a higher predictive value of MTV and TLG in comparison to the one of SUV_max_. However, no value of prediction was shown for central LNM, extrathyroidal extension, or tumoral multifocality.

Kim S.J. et al. [22] evaluated the predictive value of intratumoral heterogeneity of [^18^F]FDG uptake, correlated to PET/CT volumetric parameters, for characterization of [^18^F]FDG thyroid incidentaloma in 200 out of 493 patients who were cytologically confirmed with a thyroid nodule and enclosed in one of the Bethesda categories. The study showed no statistical significance for the PET/CT volumetric parameters, but stated that a cut-off value over 2.751 for the heterogeneity factor (HF > 2.751) had a sensitivity and specificity of 100% and 60% and could differentiate the malignant from the benign thyroid nodules.

Shi et al. [23] evaluated 99 patients with hypermetabolic thyroid incidentalomas on [^18^F]FDG PET/CT and discovered a malignancy rate of 64.6%. Their results showed that semi-quantitative PET/CT parameters were different between the malignant and the benign thyroid nodules, with a threshold SUV_max_ of 4.45 that presented a 90.6% sensitivity in distinguishing between the two types of nodules. The study also revealed that MTV and TLG indexes were significantly higher in malignant incidentalomas than in benign ones, stating that MTV 4.0 and TLG 4.0 had the greatest performance for differentiating between the FDG-avid thyroid nodules.

The same author [24] later tried to correlate volume-based [^18^F]FDG PET/CT parameters with diffusion-weighted magnetic resonance imaging (DWI) and apparent diffusion coefficient (ADC) values and ultrasound elastography (USE), in an attempt to evaluate the diagnostic performance of multi-modality functional imaging in the differential diagnosis of thyroid nodules. The group evaluated 113 patients with FDG avid thyroid incidentaloma, out of which 87 were included in this study. The parameters which showed relevance as markers of malignancy were TLG ≥ 2.475, ADC ≤ 1.8 × 10^−3^ mm^2^/s and USE score of 4. The PET/CT volumetric parameters, although consistently higher in the malignant nodules than in the benign ones, did not reach statistical significance.

Thuillier et al. [25] investigated the predictive value of different PET/CT quantitative parameters and a thyroid-adapted Deauville 5-point scale (DS) in diagnosis of malignant thyroid incidentalomas. Forty-one out of ninety-two thyroid incidentalomas were classified as benign or malignant according to cytological and histological data and were included in the study. The results showed a higher SUV_max_ value in malignant nodules than in benign ones, but without statistical significance. The group also evaluated the tumour-to-liver ratio (TLR), tumour-to-blood-pool ratio (TBR), MTV, and TLG, which had higher values in the malignant group, but without statistical relevance. The same result has been found for the Deauville-like 5-point scale approached in this study.

Erdogan et al. [26] aimed to assess the roles of MTV and TLG in predicting the malignancy risk of incidental thyroid nodules detected by [^18^F]FDG PET/CT in 101 patients. The results revealed that SUV_max_ (at a 2.4 cut-off value) and TLG could be useful in determining the risk of malignancy in nodules with FNAB indication based on EU-TIRADS risk classification in thyroid nodules detected by PET/CT. Although the MTV values of malignant nodules were higher than in the benign ones, it was statistically insignificant.

The studies that investigated the radiomics features of [^18^F]FDG PET/CT also evaluated quantitative volumetric parameters. Sollini et al. [13] identified SUV_std_, SUV_max_, MTV, and TLG as potential predictors of malignancy, correlated to radiomics features. Ceriani et al. [27] evaluated SUV_std_, SUV_peak_, SUV_max_, MTV, and TLG and discovered that such parameters reliably predict the final diagnosis on univariate analysis. TLG was discovered to be the best predictor, correctly distinguishing 79% of lesions. Aksu et al. [28] investigated six conventional parameters in their study, excluding MTV, but chose to include only SUV_max_ values in their predictive model. Giovanella et al. [29] estimated the values of SUV_max_, SUV_mean_, MTV, and TLG, but only SUV_max_ and TLG retained statistical significance as malignancy predictors. However, these studies did not focus on the usage of volumetric parameters as predictors for malignancy in FDG-avid thyroid incidentalomas, but on their integration in further predictive models, alongside histogram-based or texture radiomics features.

In this review, we chose to perform an analysis on the SUV_max_ stated by each of the articles included. Table 1 summarizes the mean ± SD of the PET/CT volumetric parameters discovered by each study group [13,20,21,22,23,24,25,26,27,28,29]. It is to be noted that authors Giovanella et al. [29] did not provide any values for either SUV_max_ or other volumetric parameters, so our pooled analysis is based on the remaining 10 studies included in this review.

A total of 334 malignant lesions were identified in the eligible papers and included in our analysis. The mean maximum standard uptake value for the malignant lesions was 9.85 ± 3.09, with a minimum value of 5.33 ± 3.93 (95% CI = 7.64–12.07), stated by Erdogan et al. [26], and a maximum SUV_max_ value of 16.11 ± 38.99 estimated from Aksu et al. [28]; however, this value did not show any statistical significance (*p* = 0.994). Our pooled analysis estimated mean MTV and TLG values for malignant lesions equal to 7.42 ± 8.08, respectively 70.82 ± 93.62, but these results did not show any statistical significance either (*p* > 0.05 in both cases), as seen in Table 2. 

### 3.2. The Potential Value of Radiomics as a Diagnostic Tool in [^18^F]FDG-Avid Thyroid Incidentalomas

A total of four papers fulfilled the inclusion criteria for radiomics-based studies and were, therefore, selected for further analysis (Table 3).

Sollini et al. [13] analysed the role of textural features in stratifying 50 patients with thyroid incidentaloma identified on [^18^F]FDG PET/CT (mean age 63 ± 15 years), that were cytologically confirmed through FNAB and grouped in five categories according to SIAPEC-IAP 2007 classification. Regions of interest were extracted using a threshold of 40% SUV_max_ and the dataset was analysed using the LifeX software package. Out of the total of 43 features extracted, 7 features were identified as potential predictors: SUV_std_ and SUV_max_, which showed the highest specificity, MTV, TLG, skewness, kurtosis and Correlation grey-level co-occurrence matrix (Correlation_GLCM_), which resulted in better sensitivity. Among the potential predictors, a mutual correlation was described between SUV_std_ and SUV_max_ (AUC 0.967), TLG and MTV (AUC 0.970), and skewness and kurtosis (AUC 0.830). In this study, skewness was the only textural feature that presented potential in differentiating benign and malignant nodules. A high negative predictive value (NPV) was also shown by Correlation_GLCM_, making it useful in ruling out the diagnosis of malignant nodules, although with a low specificity. However, the study did not propose a diagnostic predictive model and validation of this study’s results has not been performed.

Ceriani et al. [27] evaluated the ability of PET metrics and radiomics features to predict the final diagnosis of [^18^F]FDG-avid thyroid incidentalomas in 107 nodules from 104 patients (median age 65 years). Reconstructed images were analysed with PyRadiomics software package version 2.2.0, and 107 standardized features were extracted from the segmented volumes: 14 shape-based features, 18 first-order features, and 75 matrix-based features (24 GLCM features, 16 grey-level run length-matrix (GLRLM), 16 grey-level size zone matrix (GLSZM), 5 neighbouring grey tone difference matrix (NGTDM), and 14 grey-level dependence matrix (GLDM) features). Malignant nodules were confirmed by surgical pathology examination. Among the functional and volume-based PET/CT parameters, SUV_max_, SUV_peak_, SUV_mean_, TLG and MTV were significantly higher in the malignant group, compared to the benign one. TLG was the best predictor, low values being associated with benign lesions, with a NPV of 84%, while higher values had a positive predictive value (PPV) of 65% for malignant nodules.

Among the 107 radiomics features extracted, the correlation-based feature selection algorithm selected 6 non-redundant and uncorrelated radiomics features as potential predictors of malignancy: shape_Sphericity, shape_Maximum 2D DimetersSlice, firstorder_Energy, GLCM_contrast, GLCM_Inverse Difference Moment, GLCM_SumSquares. In a multivariate stepwise logistic regression analysis that included both the PET metrics and these six radiomics features, TLG, SUV_max_ and shape_Sphericity retained statistical significance (*p* < 0.0001). Based on these results, the group proposed a multiparametric predictive model integrating TLG, SUVmax, and shape_Sphericity, which was accurate in stratifying the risk of malignancy in thyroid incidentalomas (all triple positive nodules were malignant, with a PPV of 100%).

In a study by Aksu et al. [28], the authors investigated the ability of [^18^F]FDG PET/CT texture analysis to predict the exact pathological outcome of thyroid incidentalomas in 57 oncological patients and 3 non-oncological patients, who also underwent ultrasound-guided FNAB. The particularity of this study is that the authors divided the patients into two sets: a train set, which included 42 patients (mean age 63.5 ± 14.3 years), and a test set, consisting of 18 patients (mean age 65.7 ± 12.1 years. The analysis of the train set concluded that all the conventional parameters, 5 first-order features and 16 s-order features were significantly different between benign and malignant nodules, the highest benign-malignant discriminating power being found for grey-level run length matrix—run length non-uniformity (GLRLMRLNU), with an NPV of 100% and a median value of 60.7. After performing correlation analysis on the 18 features that had area under the curve (AUC) above 0.7, the parameters that had a correlation coefficient of less than 0.6 were GLRLMRLNU and SUVmax, and were then evaluated to build a predictive model. Random forest algorithm showed the best model accuracy and the highest AUC (0.849), distinguishing between the malignant and benign thyroid incidentalomas with an accuracy of 78.6%.

Giovanella et al. [29] performed a study to investigate whether radiomics analysis could improve the [^18^F]FDG PET/CT-based risk stratification in cytologically indeterminate thyroid nodules in 78 patients with an [^18^F]FDG-avid thyroid nodule that presented an indeterminate cytological report (according to the Bethesda system), but with a final histological diagnosis. The radiomics features were extracted from the reconstructed PET/CT images using PyRadiomcs software package version 2.2.0, resulting in 107 standardized features: 14 shaped-based features, 18 first-order statistics features, and 75 matrix-based features (24 GLCM, 16 GLRLM, 16 GLSZM, 5 NGTDM, and 14 GLDM). TSH levels were added to the volumetric parameters and the radiomics features, as an additional tool for decision-making. After excluding the highly correlated parameters and performing the least absolute shrinkage and selection operator (LASSO) logistic regression, two non-redundant predictors of malignancy were identified: shape_Sphericity and GLCM_Autocorrelation (AUC 0.733). The predictive model was tested both with the inclusion of patients carrying Hürthle cell adenomas and after excluding them from the patient lot, the multivariate analysis confirming that an increased radiomics score added to Bethesda class IV can predict the final diagnosis of thyroid cancer, even when Hürthle cell lesions are included.

All software used in these papers were in conformity with The Image Biomarker Standardization Initiative (IBSI) guidelines [30]. To assess the overall quality of the considered radiomics features, we adopted the radiomics quality score RQS metric described by Lambin and colleagues in 2017 [19]. This score is based on sixteen different aspects that can be grouped in six main categories: (1) protocol quality and stability in image and segmentation, (2) feature selection and validation, (3) biologic/clinical validation and utility, (4) model performance index, (5) high level evidence, and (6) open science data. The maximum value of RQS is 36. Figure 2 shows the completion rate of each RQS task for the four studies included in our review. The radiomic quality score ranged from 8 (25%) for Sollini et al. [13] to 11 (30.55%) for each of the other three studies—Ceriani et al [27], Aksu et al. [28], and Giovanella et al. [29].

## 4. Discussion

Thyroid incidentalomas are a relatively frequent finding on PET/CT studies. At the present moment, performing both dedicated thyroid ultrasound and fine-needle aspiration biopsy is considered to be the “golden standard” to rule out the diagnosis of cancer in thyroid incidentalomas discovered on [^18^F]FDG PET/CT [31]. The tumoral heterogeneity might represent a limit for the use of FNAB, as there may be differences in properties as the growth rate, vascularity, and necrosis within the same tumour cell population, outlining that between 11–42% FNABs are reported as indeterminate [22]. However, this feature raises a huge potential for radiomics on non-invasive imaging, extracting image features that, combined with quantification of tumoral volume-based PET/CT parameters, could lead to an accurate diagnosis without the need for invasive procedures [32]. Our review is based on a qualitative synthesis of eleven studies which analysed the prognostic value of PET/CT volumetric parameters in detection of malignant [^18^F]FDG-avid thyroid incidentalomas, four of which also included radiomics in their study and proposed a predictive model.

All the papers included in this review were based on retrospective studies. Although closer to the ones found in literature, we noticed that the values for the malignancy rate are very heterogenous. Most of the papers which included a higher number of patients in their cohort revealed malignancy rate from 9.8% [26] to 28% [27], presenting an incidence ranging from 15 malignant lesions/153 nodules, for Erdogan et al. [26], to 30 malignant nodules/107 for Ceriani et al. [27]. On the other hand, we noticed that the studies with the highest rate of malignancy (64.6%, respectively 59.8%) are written by the same group of authors (Shie et al. [23,24]), who stated that some of the limitations for their studies include the patient confirmation by FNAB, which might mimic other disease and be dependent on the experience of the investigator and cytopathologist that performed the examination, in addition to having a small study cohort. The other papers that presented a relatively high malignancy rate were Sollini et al. [13] (36%) and Aksu et al. [28] (44.7%), which included radiomics analysis and were based on a limited number of patients, this being an important factor when considering the estimation of results. Finally, the prevalence of malignancy on thyroid focal uptake in Asian populations is higher than in the European and American ones [9], and differences between the Asian/Eastern and Western studies included in our review have been noted.

All authors included in our review stated that the most frequent malignant histological type of thyroid incidentaloma was papillary thyroid carcinoma. There is not much research data on the topic of how the histological type of a nodule might correlate to the uptake of the [^18^F]FDG, as the papers that studied this radiotracer’s uptake on incidental thyroid findings reinforce the statement that a FNAB is needed for a correct discrimination between benign and malignant focal uptakes. One review, by Tsubaki et al. [33], states that malignant lesions that overexpress GLUT-1 on the cell membranes show a higher uptake of [^18^F]FDG and are correlated with a more aggressive histotype. The same author declares that, usually, well differentiated thyroid cancer has a low degree of [^18^F]FDG uptake, a higher uptake in this type of lesion raising the suspicion of perithyroid or lymphovascular invasion. Another review, by Soelberg et al. [9], affirms that focal lesions with a diffuse increase in surrounding thyroid uptake or a very low attenuation on CT were benign on FNAB. Aksu et al. [28] discovered that Hürthle cell adenomas generally have a higher radiotracer uptake than other benign lesions. There is little data about the way volumetric parameters and radiomics might correlate with the histologic type of the tumour, as this topic is still under-researched at present times.

The volumetric parameters have demonstrated great potential in clinical staging, evaluation of therapeutic response, and prognosis of patients with various types of cancers [34,35,36]. Because they reflect the metabolic information in the entire tumour, they can evaluate the tumoral characteristics with greater accuracy rather than a single voxel measure, as in the case of SUV_max_ [37]. Furthermore, more studies have shown that the use of just SUV_max_ as a prognostic parameter does not possess statistical significance.

SUV is a semiquantitative parameter that mirrors the metabolic activity, and large series of studies and reviews have tried to evaluate its importance as a predictor of malignancy in [^18^F]FDG-avid thyroid incidentalomas and establish a cut-off value. Systematic reviews made by Soelberg et al. [9], Shie et al. [38], and Qu et al. [39] state that, in general, malignant thyroid incidentalomas have a higher intensity of [^18^F]FDG uptake compared to the benign ones, but a statistically significant SUV_max_ cut-off value could not be established, because of the values overlap between the malignant and benign groups.

One of the biggest retrospective studies conducted by Bertagna et al. [40] in three Italian Nuclear Medicine Centres evaluated 729 thyroid incidentalomas in 49,519 patients who underwent [^18^F]FDG PET/CT for oncologic purposes and included 211 lesions in their studies. They stated a malignancy rate of 34.1%, but obtained different average SUV_max_ and cut-off values between the centres: the first centre discovered a mean SUV_max_ for malignant lesions of 15.4 ± 14.4 at a cut-off value of 4.8; the second centre stated an average SUV_max_ of 6.8 ± 1.9 for malignant nodules, and a cut-off value of 5.3; the third centre established a mean SUV_max_ value of 13.3 ± 15.3 at a SUV_max_ cut-off value of 7. The author states that there is a significant difference in [^18^F]FDG uptake between the benign and the malignant group, but a diagnostic and statistically significant cut-off value for SUV_max_ could not be demonstrated.

In our review, the authors that showed statistical significance for their discovered values of SUV_max_ in [^18^F]FDG-avid thyroid incidentalomas were Kim B.H. et al. [20,21], Shi et al. [23], Erdogan et al. [26], Sollini et al. [13], Ceriani et al. [27], Aksu et al. [28], and Giovanella et al.[29]. The author of [29] included SUV_max_ as a potential predictor of malignancy in their radiomics studies. However, given the fact that a standardized cut-off value for these parameters has not been officially stated and knowing the possibility of a vast overlap between the malignant and benign values, as stated by the aforementioned studies, SUV_max_ as a stand-alone prognostic parameters is not recommended. Therefore, the need for volumetric parameters assessment has arisen.

Volume-based PET/CT parameters, such as MTV and TLG, have been developed to enable measuring the metabolic activity in the entire tumoral mass. MTV is a volumetric measurement of tumour cells with high glycolytic activity, while TLG is defined as the product of MTV and SUV. With the development of software capable of automated volume-of-interest (VOI) delineation, the assessment of volumetric parameters has become widely used in quantitative PET [41].

Our review unravelled that TLG could be a better predictor of malignancy than MTV, as stated by Erdogan et al. [26] and Giovanella et al. [29]; however, some papers stated that their values of volume-based parameters did not reach statistical significance: Kim S.J. et al. [22], Shi et al. [24], and Thuillier et al. [25]. Kim B.H. et al. [21] stated that volumetric parameters measured by [^18^F]FDG PET/CT were significantly associated with lateral LNM in patients with incidentally detected malignant nodules, but no value of prediction was shown for central LNM, extrathyroidal extension, or tumoral multifocality. Two papers, by Kim B.H. et al. [21] and Shi et al. [23], showed that only TLG and MTV at a specific cut-off value for SUV_max_ (MTV4 and TLG4) could be used as predictors of malignancy in thyroid incidentalomas. Our pooled analysis for MTV and TLG values did not reach any statistical significance.

Orlhac et al. [14] analysed the relationship between texture parameters, histogram indices, and volume-based parameters in a study that included three patient groups that suffered from metastatic colorectal cancer, non-small cell lung cancer, and breast cancer that underwent [^18^F]FDG PET/CT examinations. Their results showed that histogram-based features are more likely to be correlated with SUV values, whereas some texture features are correlated to MTV, whatever the tumour type. The study group proposed a resampling formula with at least 32 grey levels that should avoid misleading relationships between texture indices and SUV, which should be accounted when interpreting the tumour characterization. Their study is used as a precursor to modern-day radiomics.

One of the main targets of radiomics is assessing intratumoral heterogeneity, which aims to build image-based predictive models for better patient management. The workflow of radiomics implies four main steps: (1) image acquisition and segmentation; (2) image processing; (3) feature extraction, and (4) feature selection/dimension reduction, each step affecting the final result [42].

The opportunity to correctly identify and diagnose thyroid incidentalomas can influence further investigations, therapy, and overall survival and quality of life of patients, considering the fact that most of the patients that undergo [^18^F]FDG PET/CT suffer from another non-thyroid baseline disease. The prediction models proposed by Ceriani et al. [27], Aksu et al. [28], and Giovanella et al. [29] could represent an important step in the management of this type of incidentalomas, if further investigated. The added value of radiomics features in the development of computer-aided diagnosis (CAD) systems is that these types of features are quantitative objective features automatically extracted from images [15]. Up until now, CAD systems have proven useful in the radiological field, for the diagnosis of breast cancer and lung tumours [43,44]. In nuclear medicine, deep learning has been successfully tested for the detection of bone metastases and cardiovascular disease on myocardial perfusion images [45,46].

There are, unfortunately, several limitations to the use of radiomics in general practice. First of all, radiomics is still population-dependent, meaning that acquisition devices, protocol, and results differ between the study groups, as a standardized method has not been implemented yet, thus influencing generalizability. Therefore, the majority of studies lack external validation and need further evaluation. Secondly, although radiomics should at least partially reflect the molecular biological level, variations in tumour cells and the superficial explanation given to each radiomic feature hinders data interpretability and restricts the interpretation of the final result. Moreover, the accuracy of predictive models in diagnosing malignant nodules is questionable without considering the influence of clinical information, such as tumoral staging. Finally, the ethical implications of using radiomics as a diagnostic mean and outcome predictor should be treated with caution, as this field still needs extensive research until proven safe. 

## 5. Conclusions

Despite the few articles published in literature so far, the use of volume-based PET/CT parameters and radiomics analysis in patients with thyroid incidentalomas on [^18^F]FDG PET/CT examinations outlines a great prospect in diagnosis and stratification of patients with malignant nodules and may represent a way of limiting the need for unnecessary invasive procedures, such as fine-needle aspiration biopsy or surgical excision of the thyroid.

Nevertheless, to confirm the usefulness of radiomics in the management of patients with incidental thyroid findings on [^18^F]FDG PET/CT, further studies need to be performed and external validation of the present studies needs to be made, in order to obtain a standardized protocol that can be applied without depending on the used software or the patient lot characteristics, hence making the results reproductible.

## Figures and Tables

**Figure 1 diagnostics-12-00471-f001:**
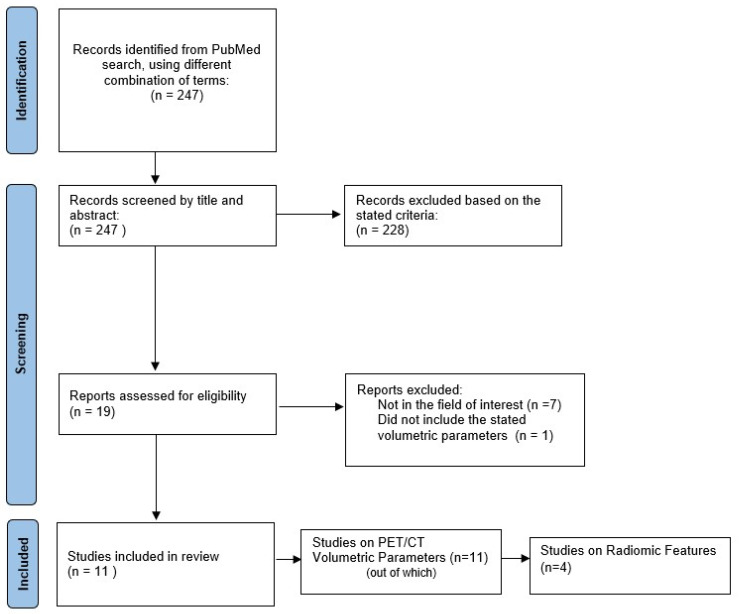
Schematic representation of the process of selection of literature data including in this qualitative review.

**Figure 2 diagnostics-12-00471-f002:**
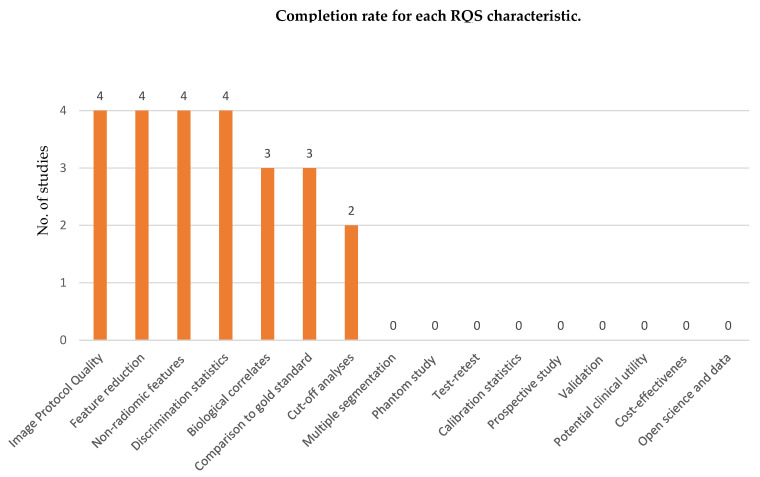
Completion rate of each radiomics quality score characteristic for the 4 studies included in this review.

**Table 1 diagnostics-12-00471-t001:** Mean and standard deviation for PET-based volumetric parameters in the malignant group.

Reference	Malignancy Rate	Mean SUVmax (Mean ± SD)	Mean MTV (Mean ± SD)	Mean TLG (Mean ± SD)
Kim B.H. et al. [20]	20.9%	8.27 ± 11.09 (AUC: 0.601)	0.27 ± 0.39 (AUC: 0.613)	NA
Kim B.H. et al. [21]	21.7%	10.52 ± 15.28 (AUC: 0.716)	4.18 ± 7.68 (AUC: 0.839)	49.33 ± 599.39 (AUC: 0.815)
Kim S.J. et al. [22]	NA	5.96 ± 2.61 (AUC: 0.586)	5.76 ± 2.0 (AUC: 0.566)	16.01 ± 6.9 (AUC: 0.562)
Shi et al. [23]	64.6%	11.30 ± 8.40 (AUC: 0.866)	2.7 ± 4.0 (AUC: 0.872)	30.0 ± 75.5 (AUC: 0.895)
Shi et al. [24]	59.8%	11.90 ± 8.90 (AUC: 0.872)	3.06 ± 4.30 (AUC: 0.895)	35.2 ± 83.0 (AUC: 0.916)
Thuillier et al. [25]	NA	9.27 ± 3.28 (AUC: 0.550)	5.46 ± 12.18 (AUC: 0.530)	22.66 ± 38.41 (AUC: 0.610)
Erdogan et al. [26]	9.8%	5.33 ± 2.93 (AUC: 0.827)	5.76 ± 9.78 (AUC: 0.668)	21.14 ± 37.51 (AUC: 0.726)
Sollini et al. [13]	36%	9 ± 8.70 (AUC: 0.600)	27 ± 94.5 (AUC: 0.660)	309.5 ± 1881.7 (AUC: 0.660)
Ceriani et al. [27]	28%	10.91 ± 2.04 (AUC: 0.652)	12.60 ± 7.46 (AUC: 0.733)	45.33 ± 13.84 (AUC: 0.756)
Aksu et al. [28]	44.7%	16.11 ± 38.99 (AUC: 0.758)	NA	107.59 ± 213.16 (AUC: 0.822)

Abbreviations: SUV—standardized uptake value; MTV—Metabolic Tumour Volume; TLG—Total Lesion Glycolysis; SD—Standard Deviation; NA—not applicable.

**Table 2 diagnostics-12-00471-t002:** Volumetric parameters pooled values obtained in our sub-analysis.

PET/CT Parameter	Mean SUVmax (Mean ± SD)	Mean MTV (Mean ± SD)	Mean TLG (Mean ± SD)
Obtained Pooled Value	9.85 ± 3.09	7.42 ± 8.08	70.82 ± 93.62
*p*-value	*p* = 0.994	*p* = 1	*p* = 1

**Table 3 diagnostics-12-00471-t003:** Studies that used radiomics to differentiate malignant from benign thyroid incidentalomas.

Reference	No. Patients/Lesions	ROI Segmentation	Software	No. Radiomics Features	Model Construction	Components of Predictive Model	Validation Method	AUC
Sollini et al. [13]	50	Fixed threshold (SUV_max_ > 40%)	LifeX package	43	NA	NA	NA	NA
Ceriani et al. [27]	107 nodules (104 patients)	Fixed threshold	PyRadiomics Version 2.2.0	107	Univariate Logistic Regression of Dichotomized Data Logistic Stepwise Regression Function	SUV_max_ TLG Shape_Sphericity	1000-resampled bootstrapping CV	0.830
Aksu et al. [28]	60 (42 train set, 18 test set)	Fixed threshold (SUV_max_ > 40%)	LifeX package	46	RF SVM DT NB *k* nearest neighbour Logistic regression for binary risk classification	SUV_max_ GLRLM_RLNU_	Tenfold CV + EV	0.849
Giovanella et al. [29]	78	Fixed threshold algorithm (mean SUV of the contralateral lobe)	PyRadiomics Version 2.2.0	107	LASSO (with tenfold CV)	Shape_Sphericity GLCM_autocorrelation	1000-resampled bootstrapping CV	0.733

Abbreviations: ROI—region of interest; AUC—area under curve; SUV—standardized uptake value; TLG—total lesion glycolysis; NA—not applicable; CV—cross-validation; RF—random forest; SVM—support vector machine; DT—decision tree; NB—naïve bayes; LASSO—least absolute shrinkage and selection operator; GLRLM—grey-level run length matrix; GLCM—grey-level co-occurrence matrix; EV—external validation.

## Data Availability

Not applicable.

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
