# Peer review of "Radiomics Analysis of [18F]FDG PET/CT Thyroid Incidentalomas: How Can It Improve Patients’ Clinical Management? A Systematic Review from the Literature"

_diagnostics, 2022, doi:10.3390/diagnostics12020471_

Round 1
Reviewer 1 Report
The authors reviewed the literature about PET-based radiomics of [18F]FDG-avid thyroid incidentalomas and discussed the additional value of PET volumetric parameters and radiomic features. The topic is of interest and the manuscript is well written. However, a few remarks should be addressed:
1. In Table 1, the author listed the malignancy rate of the ten studies. We can find the malignancy rate ranged from 9.8% to 64.6%. More analysis of the huge difference among these studies is needed. How and which factor cause the difference?
2. In the present manuscript, the authors evaluated the use of PET volumetric parameters and radiomics analysis in patients with [18F]FDG-avid thyroid incidentalomas. As we know, the uptake of [18F]FDG correlates with the proliferation and differentiation of malignant tumors. Could you provide more histopathologic information of the FDG-avid thyroid incidentalomas. How do PET volumetric parameters and radiomics features correlate with the histopathologic type of thyroid incidentalomas?
3. In the section of discussion, the authors mentioned that “Thyroid incidentalomas are a relatively frequent finding on PET/CT studies, not only when using [18F]FDG, but in the ones performed using other radiotracers as well, such as [18F]Choline and [68Ga]PSMA”. Could the authors comment on the PET volumetric parameters and radiomics analysis for other tracers, such as PSMA, etc?
Author Response
Answers
In Table 1, the author listed the malignancy rate of the ten studies. We can find the malignancy rate ranged from 9.8% to 64.6%. More analysis of the huge difference among these studies is needed. How and which factor cause the difference?
We have analysed your suggestions and inserted the below paragraph between lines 381-402, page 10-11.
All the papers included in this review were based on retrospective studies. Although closer to the ones found in literature, we noticed that the values for the malignancy rate are very heterogenous. Most of the papers which included a higher number of patients in their cohort revealed malignancy rate from 9.8% [24] to 28%[26], presenting an incidence ranging from 15 malignant lesions/153 nodules, for Erdogan et al.[24], to 30 malignant nodules/107 for Ceriani et al.[26]. On the other hand, we noticed that the studies with the highest rate of malignancy (64.6%, respectively 59.8%) are written by the same group of authors (Shie et al. [21,22]), who stated that some of the limitations for their studies include the patient confirmation by FNAB, which might mimic other disease and be dependent on the experience of the investigator and cytopathologist that performed the examination, in addition to having a small study cohort. The other papers that presented a relatively high malignancy rate were Sollini et al.[25] (36%) and Aksu et al.[27] (44.7%), which included radiomics analysis and were based on a limited number of patients, this being an important factor when considering the estimation of results. Finally, the prevalence of malignancy on thyroid focal uptake in Asian populations is higher than in the European and American ones [9], and differences between the Asian/Eastern and Western studies included in our review have been noted.
- In the present manuscript, the authors evaluated the use of PET volumetric parameters and radiomics analysis in patients with [18F]FDG-avid thyroid incidentalomas. As we know, the uptake of [18F]FDG correlates with the proliferation and differentiation of malignant tumors. Could you provide more histopathologic information of the FDG-avid thyroid incidentalomas. How do PET volumetric parameters and radiomics features correlate with the histopathologic type of thyroid incidentalomas?
We have analysed your suggestions and inserted the below paragraph between lines 403-418, page 11.
All authors included in our review stated that the most frequent malignant histological type of thyroid incidentaloma was papillary thyroid carcinoma. There is not much research data on the topic of how the histological type of a nodule might correlate to the uptake of the [18F]FDG, as the papers that studied this radiotracer’s uptake on incidental thyroid findings reinforce the statement that a FNAB is needed for a correct discrimination between benign and malignant focal uptakes. One review, by Tsubaki et al.[35], states that malignant lesions that overexpress GLUT-1 on the cell membranes show a higher uptake of [18F]FDG and are correlated with a more aggressive histotype. The same author declares that, usually, well differentiated thyroid cancer has a low degree of [18F]FDG uptake, a higher uptake in this type of lesion raising the suspicion of perithyroid or lymphovascular invasion. Another review, by Soelberg et al. [9], affirms that focal lesions with a diffuse increase in surrounding thyroid uptake or a very low attenuation on CT were benign on FNAB. Aksu et al. [27] discovered that Hürthle cell adenomas generally have a higher radiotracer uptake than other benign lesions. There is little data about the way volumetric parameters and radiomics might correlate with the histologic type of the tumour, as this topic is still underresearched at present times.
- In the section of discussion, the authors mentioned that “Thyroid incidentalomas are a relatively frequent finding on PET/CT studies, not only when using [18F]FDG, but in the ones performed using other radiotracers as well, such as [18F]Choline and [68Ga]PSMA”. Could the authors comment on the PET volumetric parameters and radiomics analysis for other tracers, such as PSMA, etc?
[18F]Choline and [68Ga]PSMA -avid thyroid incidentalomas are understudied (there are few papers that comment on their appearance on PET/CT studies), therefore there are no articles about volumetric parameters or radiomics analysis on this matter. We have erased this phrase.
Reviewer 2 Report
The article is well written, and the conclusions clear. However,
- The clinical impact of using artificial intelligence to replace FNAB seems not very realistic.
Indeed, FNAB is a minimal invasive method, which should not be compared to surgical biopsy (page 2, lines 47-48). FNAB is a well-established methodology, which provides fundamental information for the management of patients in term of prognosis and therapy. The statement that “the tumoral heterogeneity limits the use of biopsy” (page 310, line 320) should be better supported, if not canceled.
Conversely, the evaluation of AI currently comes up against several methodological difficulties (illustrated in the text by the authors). Given that “between 14-63.3% (of incidentalomas) have been confirmed as malignant”, the risk of underestimating a potentially aggressive tumour (high FDG uptake) is too high to avoid FNAB.
- I strongly suggest erasing the analysis on the SUVmax performed by assembling the data from the included articles (page 65, lines 207-219).
The authors acknowledged that “these results did not show any statistical significance”. Moreover, without a proper standardization of the measurements, the values of SUVmax evaluated on different PET/CT are not comparable. This is recognized by the authors themselves, presenting an “overlap between the malignant and benign groups”.
Minor comments
page 1 line 36: please replace “imagistic” by “imaging”
page 1 lines 44-45: please clarify the sentence “by approximatively one third of the patients who underwent further investigation”
Table 1: please add statistical data to each row.
Author Response
Answers
- The clinical impact of using artificial intelligence to replace FNAB seems not very realistic.
Indeed, FNAB is a minimal invasive method, which should not be compared to surgical biopsy (page 2, lines 47-48). FNAB is a well-established methodology, which provides fundamental information for the management of patients in term of prognosis and therapy. The statement that “the tumoral heterogeneity limits the use of biopsy” (page 310, line 320) should be better supported, if not canceled.
Conversely, the evaluation of AI currently comes up against several methodological difficulties (illustrated in the text by the authors). Given that “between 14-63.3% (of incidentalomas) have been confirmed as malignant”, the risk of underestimating a potentially aggressive tumour (high FDG uptake) is too high to avoid FNAB.
We have changed the statements in question according to your suggestion, by rephrasing the sentences in a more clear way and providing further explanation where necessary. (lines 52-53, lines 371-374).
- I strongly suggest erasing the analysis on the SUVmax performed by assembling the data from the included articles (page 65, lines 207-219).
The authors acknowledged that “these results did not show any statistical significance”. Moreover, without a proper standardization of the measurements, the values of SUVmax evaluated on different PET/CT are not comparable. This is recognized by the authors themselves, presenting an “overlap between the malignant and benign groups”.
We believe erasing the analysis of the SUVmax would eliminate an important piece of information from our subanalysis, as it is one of the important parameters to measure when analyzing PET-CT studies and, together with MTV, aids in the calculation of TLG. In addition to this, SUVmax is used as a main parameter for assembling a predictive model in two of the papers based on radiomics. We have stated that neither of our obtained values presents statistical significance due to the small amount of data included in our review and the fact that even some of the SUVmax values reported by the studies are stated to not be statistically significant, and we acknowledged the possibility of overlapping values between the malignant and benign groups and the heterogeneity of the values to highlight that SUVmax alone cannot be used as a predictor for malignity, but should be interpreted into clinical context and correlated to other volumetric and radiomic parameters. However, if after these considerations, this will still represent an issue, we will act accordingly and change the manuscript.
Minor comments
page 1 line 36: please replace “imagistic” by “imaging”
The correction has been made.
page 1 lines 44-45: please clarify the sentence “by approximatively one third of the patients who underwent further investigation”
The sentence has been reformulated.
Table 1: please add statistical data to each row.
Statistical data was added in the table.
Round 2
Reviewer 2 Report
The authors considered all the reviewers’ comments and criticisms and responded in a complete and correct way. The paper is now acceptable.